# Resistive Switching Memory Cell Property Improvement by Al/SrZrTiO_3_/Al/SrZrTiO_3_/ITO with Embedded Al Layer

**DOI:** 10.3390/nano12244412

**Published:** 2022-12-10

**Authors:** Ke-Jing Lee, Wei-Shao Lin, Li-Wen Wang, Hsin-Ni Lin, Yeong-Her Wang

**Affiliations:** 1Institute of Microelectronics, Department of Electrical Engineering, National Cheng-Kung University, Tainan 701, Taiwan; 2Program on Semiconductor Process Technology, Academy of Innovative Semiconductor and Sustainable Manufacturing, National Cheng-Kung University, Tainan 701, Taiwan; 3Department of Physics, National Sun Yat-sen University, Kaohsiung 804, Taiwan

**Keywords:** resistive random-access memory (RRAM), sol-gel, strontium zirconate titanium

## Abstract

The SrZrTiO_3_ (SZT) thin film prepared by sol-gel process for the insulator of resistive random-access memory (RRAM) is presented. Al was embedded in the SZT thin film to enhance the switching characteristics. Compared with the pure SZT thin-film RRAM, the RRAM with the embedded Al in SZT thin film demonstrated outstanding device parameter improvements, such as a resistance ratio higher than 10^7^, lower operation voltage (V_SET_ = −0.8 V and V_RESET_ = 2.05 V), uniform film, and device stability of more than 10^5^ s. The physical properties of the SZT thin film and the embedded-Al SZT thin-film RRAM devices were probed.

## 1. Introduction

Among various candidates for nonvolatile memories, resistive random-access memory (RRAM) attracts considerable attention owing to its advantages, such as simple metal–insulator–metal (MIM) structure, fast operation speed, low operation voltage, and low fabrication temperature. Numerous materials, such as transition metal oxides (HfO_2_, TiO_2_, and NiO), organic material (composites containing nanoparticles and polyimide), or perovskite oxides (SrZrO_3_ and SrTiO_3_), have been investigated as potential materials for RRAMs. In this study, we focused on the SrTiO_3_ material because of its high dielectric constant, low dielectric loss, tenability, high breakdown strength, low leakage current, and great film quality [1]. Numerous attempts have been exerted to further improve the properties of SrTiO_3_-based ceramics. Doping is considered an effective approach for altering their properties [2]. Reproducible resistive switching behaviors have been observed in doped perovskite oxide films [3,4], such as Mo-doped SrTiO_3_ [5], Mg-doped SrTiO_3_ [6], Nb-doped SrTiO_3_ films [7], and Ni-doped SrTiO_3_ films [3]. Several studies have indicated that Zr^4+^ in SrTiO_3_ can stabilize the charge of Ti^4+^ and suppress oxygen dissociation [1,2,8]. In addition, our previous study showed that the addition of Zr can effectively improve the surface morphology of insulators [9]. Moreover, the sol-gel process has advantages, such as low fabrication temperature, low cost, and easy adjustment of proportions. This process can be applied in numerous devices [10]. Based on the above advantages and research results, in this work, we utilized Zr in the doping process to fabricate SrZrTiO_3_ (SZT) using the sol-gel process for RRAM insulators.

Various inserted metal layers in the insulator layer, including Cu, Pt, and Ti, improve the resistive switching properties of RRAMs [11,12,13,14]. However, a relatively limited number of studies have been performed regarding the mechanism of embedded metal-based RRAM devices. In this study, the effects of Al-included SZT on the resistive switching properties of SZT thin film for RRAM applications were also investigated. For comparison, a memory device with a single-SZT active layer was also fabricated and characterized.

## 2. Materials and Methods

The solutions used in this study were prepared in reference to the previous literature, and the additives and ratios were varied based on experimental requirements [8,15,16]. The 0.5 M SZT solution was prepared in three steps (Figure 1a). First, strontium acetate (537 mg) was dissolved in glacial acetic acid (4 mL) by stirring and then heated on a 100 °C hot plate until complete dissolution (A1 solution). Titanium isopropoxide (760 µL) and zirconium n-propoxide (770 µL) were mixed with acetylacetone (470 µL) and dissolved in 2-methoxyethanol (7.74 mL) by stirring (A2 solution). Finally, A1 was slowly dropped into A2 until a completely transparent solution was obtained (mole proportion of Sr:Zr:Ti was approximately 1:1:1). After chemically cleaning the substrate, the prepared 0.5 M SZT solution was spun coated on an indium tin oxide (ITO)/glass substrate. Each layer was baked at 100 °C for 15 min to remove volatile materials. A 90 nm-thick Al film with an area of 3 mm^2^ was deposited with a shadow mask using DC magnetron sputtering as the top electrode (TE) of the MIM structure, whereas ITO served as the bottom electrode (BE). Figure 1b shows the schematic configuration of the Al/SZT/ITO structures.

The SZT/Al/SZT tri-layer structure was deposited on the ITO/glass substrate. Both SZT films were prepared through the same process above, and Al was embedded by DC magnetron sputtering. Finally, the same processes were applied for Al TEs. For comparison, we fabricated four different thicknesses of embedded Al layer (5, 18, 25, and 33 nm). Figure 1c displays the schematic configuration of embedded Al in SZT thin-film RRAM devices. Transmission electron microscopy (TEM) analysis was carried out using a JEM-2100F electron microscope. The X-ray diffraction (XRD) spectra were developed via BRUKER, D8 DISCOVER SSS Multi-Function High Power XRD using the copper Kα line with λ = 0.154060 nm. The roughness value and surface morphology of the films were calculated using atomic force microscopy (AFM, Dimension ICON with NanoScope V controller, Bruker, Billerica, MA, USA). X-ray photoelectron spectroscopy (XPS) was performed using a PHI 5000 VersaProbe. Electrical measurements were performed using an Agilent B1500 semiconductor parameter analyzer.

## 3. Results

Figure 2a presents the TEM image of the cross-sectional SZT/Al/SZT tri-layered thin films sandwiched between ITO BE and Al TE. The estimated thicknesses of the top SZT layer, Al interlayer, and bottom SZT layer were 54, 18, and 67 nm, respectively. The thin films were uniform, and the interfaces were clear. Figure 2b depicts the XRD analysis of the SZT thin film. As shown in the analysis results, no evident peak was observed in the XRD spectrum. The SZT thin film prepared in this experiment had an amorphous phase. Compared with the polycrystalline phase, the amorphous phase is preferred for dielectric layer materials because the former may lead to a high-grain boundary leakage current and a rough film surface [17]. The AFM images in Figure 2c,d reveal the surface morphology of the SZT and SZT/Al/SZT tri-layered thin films. The root-mean-square roughness (R_rms_) values of SZT and the SZT/Al/SZT thin film were approximately 2.0 and 0.8 nm, respectively. The roughness of the thin films decreased due to the insertion of Al layer.

The XPS analysis was used to obtain the chemical composition of SZT thin film, as shown in Figure 3a. The atomic percentage of Sr, Zr, Ti, and O were obtained as 13%, 11%, 16%, and 60%, respectively. By fitting the peaks of Sr 3d signal can be decomposed into two peaks of Sr 3d_5/2_ and Sr 3d_3/2_, which are centered at 132.7 and 134.1 eV, respectively. The peak at 132.7 eV can be attributed to SrTiO_3_, and the peak at 134.1 eV can be attributed to SrCO_3_. By fitting the peaks of Zr3d signal can be decomposed into two peaks of Zr3d5/2 and Zr 3d_3/2_, which are centered at 182 and 184.4 eV, respectively. This result confirms that most of Zr atoms are incorporated at Ti lattice site instead of forming ZrO_2_ [18,19]. The binding energy around 182 and 184.4 eV are assigned for ZrTiO_4_. The peaks for the O1s signal may be consistently fitted by three different near–Gaussian sub-peaks centered at 529.2, 530.9, and 532.1 eV. The binding energy of lattice oxygen is 529.2 eV, which is attributed to the O_2_ ions bonded with Sr, Zr, and Ti ions. The peaks located at 530.9 eV were associated with non-lattice oxygen ions, such as oxygen vacancy, and the peak at 532.1 eV corresponds to surface adsorption oxygen in the SrTiO_3_ catalyst [3,9,20]. Peaks fitting analyses of the Ti 2p signal can be decomposed into two peaks of Ti 2p_3/2_ and Ti 2p_1/2_, with two components of the binding energies at 458.8 eV and 464.3 eV, which are attributed to SrTiO_3_ and ZrTiO_4_. To investigate the possible inter-diffusion in the SZT/Al/SZT tri-layer, we determined the XPS depth profile of the SZT/Al/SZT tri-layer thin film (Figure 3b). The Al concentration increased, and Sr, Zr, and Ti concentrations correspondingly decreased with depth after 36 s Ar^+^ sputtering. This result demonstrated the significant interfacial diffusion between SZT/Al/SZT, which is in good agreement with the TEM cross-sectional image. In spite of this finding, the memory units of the SZT/Al/SZT tri-layer structure on ITO-coated glass have been fabricated by sputtering [21,22].

Figure 4a depicts the bipolar and reproducible resistive switching behaviors of the Al/SZT/ITO structures. During the negative sweep from 0 to −4 V, the current increased sharply, a phenomenon called the “set” process, switching from a high resistance state (HRS) to a low resistance state (LRS). Sequentially, during the positive sweep from 0 to 4 V, an abrupt decrease in current, that is, the “reset” process, was observed, and it switched the resistance state from LRS to HRS with 10 mA compliance current. The V_SET_/V_RESET_ of Al/SZT/ITO was −1.28 V/+2.08 V. The ON/OFF ratio of the devices was around 10^3^. Figure 4b shows the Al/SZT/SZT/ITO RRAM devices. The set voltage (V_SET_) was −2.88 V, and the reset voltage (V_RESET_) was located at +2.48 V. The ON/OFF ratio of the devices was around 10. Figure 4c–f shows the discussed resistive switching I–V characteristic of different thicknesses (5, 18, 25, and 33 nm) of the embedded Al. Therefore, the observed I–V characteristic of the 18 nm embedded Al displayed an especially outstanding performance. However, the memory margins of embedded Al layer with different thicknesses were insufficient for memory application. The V_SET_/V_RESET_ of Al/SZT/Al (18 nm)/SZT/ITO was −0.32/+1.28 V (Figure 4d). The ON/OFF ratio of the devices was around 10^7^. No significant switching behavior was observed in the embedded structure with different thicknesses (5, 25, and 33 nm) of Al (Figure 4c,e,f).

To probe the mechanisms of resistive switching characteristics of SZT-based RRAM, we performed the curve fittings of conduction mechanisms for HRS and LRS in the Al/SZT/ITO and Al/SZT/Al (18 nm)/SZT/ITO devices, and the I–V characteristic was plotted in log-log scale (Figure 5a,b). At low applied voltage, the slope of HRS was very close to linear. As the voltage increased, the current of HRS followed a voltage-square dependence. With the continuous increase in bias voltage, the current of HRS increased rapidly, corresponding to the steep increase in the current region. After the abrupt increase in its region, the current showed voltage-square dependence again. The fitting results of HRS illustrated that the current showed typical space-charge-limited conduction, which consists of the ohmic region (I ∝ V), trap-filled limit current (I ∝ V^2^), and Child’s law region (I ∝ V^2^) [23,24].

However, the conduction behaviors of the LRS showed distinct features for Al/SZT/ITO and Al/SZT/Al (18 nm)/SZT/ITO devices. By contrast, the curve fitting of LRS of Al/SZT/ITO device showed the ohmic conduction behavior, coinciding with a conducting filamentary model [25,26]. The I–V curve of the LRS of Al/SZT/Al (18 nm)/SZT/ITO device consisted of the ohmic region (I ∝ V) and trap-filled limit current (I ∝ V^2^) and Child’s law region (I ∝ V^2^), which indicates that resistance switching in Al/SZT/Al (18 nm)/SZT/ITO device was mediated by a carrier trapping/de-trapping process [27]. They can remain below V_RESET_ in LRS Part 3 but reset did not occur until V_RESET_ was reached again in HRS Part 4 because of the lag induced by the relaxation of trap-filled states (Figure 4d). The simple embedded Al layer not only remarkably improved the device parameters but essentially altered the switching mechanism.

The trap depth (ψ_t_) of ~0.91 eV for the SZT memory device at the HRS can be extrapolated from an intercept of ln(J/E) as the function of temperature (Figure 6) [24,28]. ψ_t,A_ and ψ_t,B_ are the trap depth of the Al/SZT/ITO and Al/SZT/Al (18 nm)/SZT/ITO structure, respectively. The equivalent trap depth can be reduced to ~0.44 eV after the insertion of Al layer. The possible carrier transporting according to the carriers hops along the trapping states. Under the same barrier height condition, the reduction of trap depth enhances opportunities for electron hopping, thereby significantly reducing the HRS current and leading to a high ON/OFF ratio of 10^7^. From the above results, initially, native defects of oxygen vacancies scatter in the SZT film. Injected electrons are trapped in the defects and affect the current conduction. Thus, the conduction mechanism in HRS is dominated by SCLC.

Figure 7a shows the Al/SZT/Al (18 nm)/SZT/ITO structure band diagram under zero bias. The valleys on the upper side of the SZT insulator represent the traps inside the SZT insulator. The left SZT insulator layer was leaned as the result of the different work functions between ITO and Al (ITO: 4.5 eV and Al: 4.2 eV). The Al layer embedded in the SZT thin film acted as a trap for electrons because the electron affinity of SZT was higher than the work function of Al. When the devices were under low negative bias, which was equivalent to the Al TE, a negative bias was applied, and the ITO bottom was under applied positive bias (Figure 7b). In the low applied-bias negative region, the transport in RRAM devices was caused by thermally generated carriers [11]. When the carriers crossed over the first SZT insulator from the Al TE, they were trapped inside the embedded Al layer. When the applied negative voltage increased, the traps in the SZT insulator were almost filled, the carriers began to transport in the SZT insulator, and the concentration of carriers inside the SZT insulator increased. Meanwhile, the carriers trapped in the embedded Al layer also joined the transport. In this region, the current steeply increased, and the RRAM switched from HRS to LRS. By contrast, the switch of devices from LRS to HRS was also discussed. First, when the applied positive was small, the carriers were gradually released by the trap in the SZT insulator, that is, the detrapping process (Figure 7c). When the positive arrives at the V_RESET_, the devices originally should switch from LRS to HRS. However, given the carriers trapped inside the embedded Al layer, the carrier concentration in the SZT insulator remained high, and the devices were maintained in the LRS state. During the applied positive sweeps from +4 V to the V_RESET_ (about +1.28 V), the RRAM devices reached the detrapping situation. The carriers could not fill the trap, and the carrier concentration in the SZT insulator decreased, causing the devices to switch from the LRS to HRS. As a result, the embedded Al layer structures in RRAM devices caused a decrease in the HRS current and resulted in a high on/off ratio. Further, the embedded Al layer structures in RRAM devices caused a lag in the transport process, and we can discover the phenomenon in the I–V characteristic.

Figure 8 shows the reliability issues of Al/SZT/ITO and Al/SZT/Al (18 nm)/SZT/ITO structures. The distribution of current and operation voltage was investigated. The V_SET_ and V_RESET_ distribution were plotted in a histogram to demonstrate the reliability of the Al/SZT/ITO and Al/SZT/Al (18 nm)/SZT/ITO RRAM (Figure 8a,b, respectively). In mathematics, μ is the mean, and σ is the standard deviation [29]. As presented in Figure 8a, the V_SET_ and V_RESET_ were distributed widely. The average and standard deviation values of the V_SET_ were −1.46 and 0.36, respectively. The average and standard deviation values of V_RESET_ were 2.46 and 0.36 V, respectively. As shown in Figure 8b, when the devices switched between ON and OFF states, V_SET_ was distributed in the range of −0.2 to −0.68 V, whereas V_RESET_ was distributed in the range of 0.92 to 1.52 V. The average and standard deviation values of V_SET_ were −0.44 and 0.18 V, respectively. The average and standard deviation values of V_RESET_ were 1.22 and 0.22 V, respectively. The reduction and more stable distribution of V_SET_ and V_RESET_ may be related to the addition of an 18 nm embedded Al layer [30].

Figure 8c,d illustrates the cumulative probabilities of HRS/LRS resistance for the Al/SZT/ITO and Al/SZT/Al (18 nm)/SZT/ITO device. Comparing the 18 nm embedded Al RRAM with the SZT-based RRAM, the coefficient declined from 0.26 to 0.04 at LRS and decreased by 0.03% at HRS, which reveals the 18 nm embedded Al RRAM depicted a large memory window and uniformity compared with SZT-based RRAM.

Figure 9a plots the scaling trend of the LRS resistance versus the cell area of the Al/SZT/ITO and Al/SZT/Al (18 nm)/SZT/ITO structures. The LRS resistance is mainly a filamentary conduction current, and thus, it only had a slight dependence on the cell area. These results are similar to those for other metal oxides [31]. Figure 9b presents the retention capabilities of Al/SZT/ITO and Al/SZT/Al (18 nm)/SZT/ITO structures measured at room temperature at a voltage of 0.5 V. The ON/OFF ratio of the Al/SZT/Al (18 nm)/SZT/ITO structure remained higher than 10^6^, and that for the Al/SZT/ITO structure was around 10^3^. For the Al/SZT/Al (18 nm)/SZT/ITO structure, the current magnitudes did not differ significantly over 10^5^ s. The smooth roughness may facilitate stable resistive switching [3], and the stability of the switching cycle and retention capability improved compared with those of the SZT memory device.

From Table 1, the I–V switching characteristics of RRAM devices were promoted by inserting the Al embedded layer. Compared with the pure SZT-based RRAM devices and reported data [11,12,13,14], the resistance ratio increased from 10^3^ to 10^7^, which is an ultrahigh memory margin, which led to distinguishing the storage information easily. The V_SET_ decreased from −1.28 to −0.32 V, and the V_RESET_ from 2.08 to 1.28 V. These results indicate that the operation voltage and current can be decreased to lower power consumption.

## 4. Conclusions

In conclusion, resistive switching behaviors in sol-gel SZT thin films have been probed, and the bipolar resistive switching characteristics of Al/SZT/ITO devices can utilize Al as an embedded layer in the SZT thin film for enhancing properties. Significant improvements in the ON/OFF ratios from 10^3^ to over 10^7^ were observed upon application to the Al/SZT/Al (18 nm)/SZT/ITO structure without the forming processes. A high interface trap density with a large amount of trapped electrons will help in building a favorable electric field to attract oxygen vacancies, and thus, the lower V_SET_ and V_RESET_ compared with those of the SZT memory device were obtained. The reduced trapping depths from 0.91 eV to 0.44 eV were found after the insertion of the Al layer, resulting in a low HRS current or high ON/OFF ratio. Therefore, the free-forming Al/SZT/Al (18 nm)/SZT/ITO structure with a high ON/OFF ratio of over 10^7^, excellent voltage distribution, and good retention of over 10^5^ s can be achieved.

## Figures and Tables

**Figure 1 nanomaterials-12-04412-f001:**
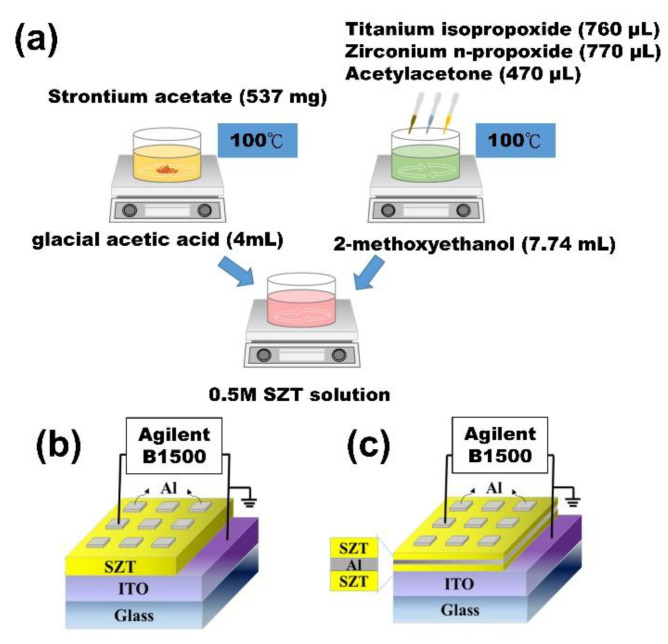
(**a**) Preparation of 0.5 M SZT solution. (**b**) Schematic of SZT-based RRAM devices. (**c**) Schematic of Al embedded in SZT thin-film RRAM devices.

**Figure 2 nanomaterials-12-04412-f002:**
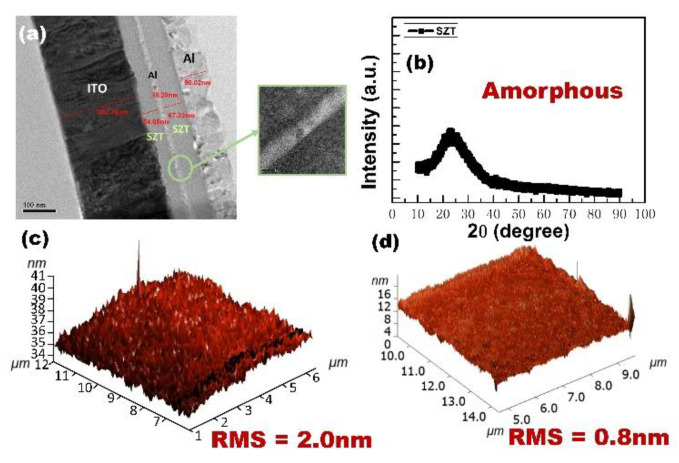
(**a**) TEM image of cross-sectional SZT/Al/SZT tri-layered thin films sandwiched between ITO BE and Al TE. (**b**) XRD spectrum of SZT thin film. AFM image of (**c**) SZT and (**d**) SZT/Al/SZT tri-layered thin films.

**Figure 3 nanomaterials-12-04412-f003:**
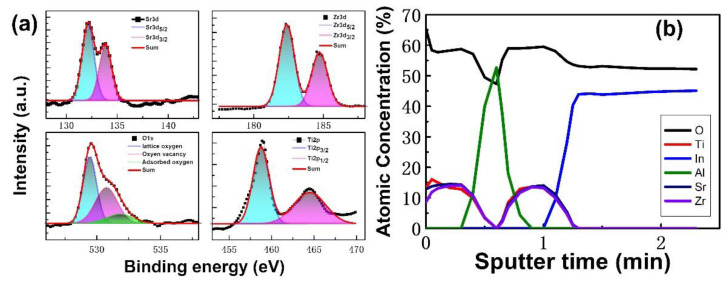
(**a**) High-resolution XPS spectra of Sr3d, Zr3d, Ti2p, and O1s of the SZT thin film. (**b**) Atomic concentrations of six elements in the SZT/Al/SZT tri-layered thin film with XPS depth profiling.

**Figure 4 nanomaterials-12-04412-f004:**
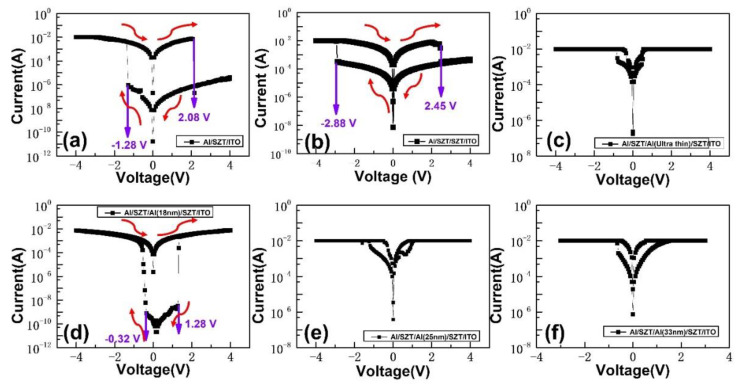
I–V switching curves of (**a**) Al/SZT/ITO, (**b**) Al/SZT/SZT/ITO, and (**c**–**f**) Al/SZT/different thicknesses (5, 18, 25, and 33 nm) of embedded metal –Al /SZT/ITO device RRAMs.

**Figure 5 nanomaterials-12-04412-f005:**
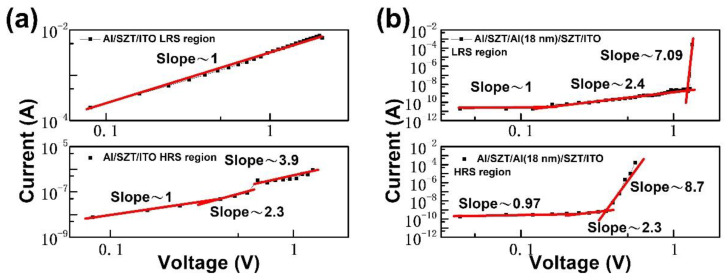
Double logarithmic plot and linear fitting of the switching I–V curve of (**a**) Al/SZT/ITO and (**b**) Al/SZT/Al (18 nm)/SZT/ITO device RRAMs.

**Figure 6 nanomaterials-12-04412-f006:**
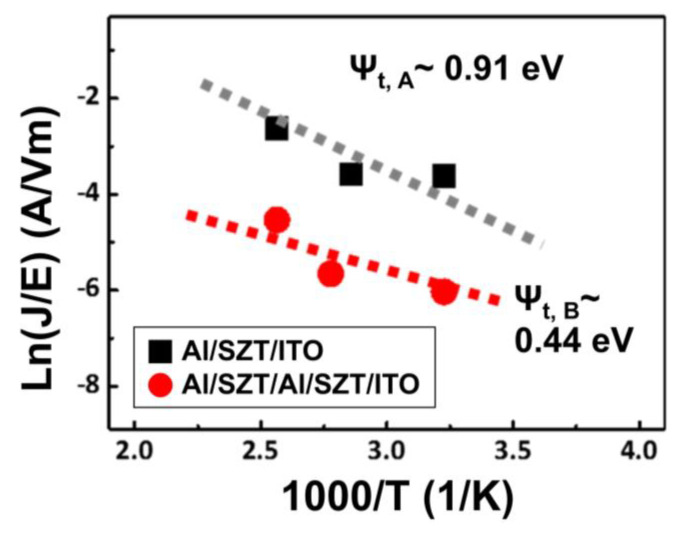
ln(J/E) as the function 1/T of the Al/SZT/ITO and Al/SZT/Al (18 nm)/SZT/ITO structure.

**Figure 7 nanomaterials-12-04412-f007:**
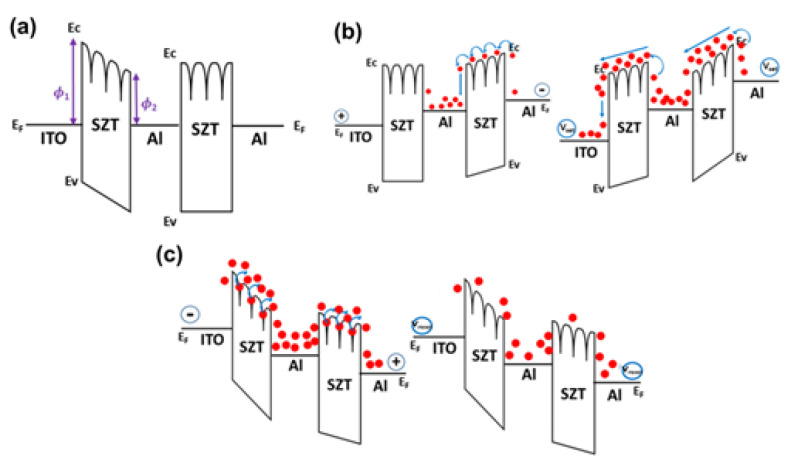
Possible resistive switching mechanism of Al/SZT/Al (18 nm)/SZT/ITO devices. (**a**) Thermal equilibrium, (**b**) negative bias, and (**c**) positive bias.

**Figure 8 nanomaterials-12-04412-f008:**
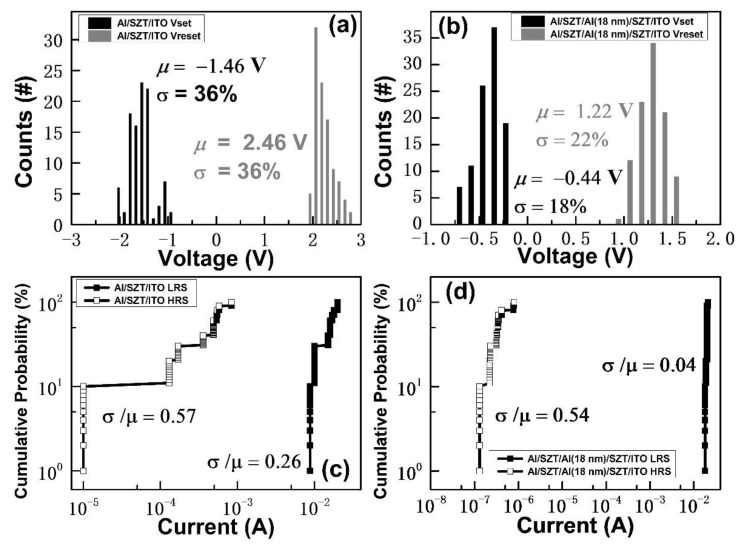
Statistical and cumulative probability distribution of V_SET_ and V_RESET_ measured from (**a**) Al/SZT/ITO and (**b**) Al/SZT/Al (18 nm)/SZT/ITO devices unit during 100 time tests. The distribution of HRS/LRS resistance of (**c**) Al/SZT/ITO and (**b**) Al/SZT/Al (18 nm)/SZT/ITO devices unit for 100 time tests.

**Figure 9 nanomaterials-12-04412-f009:**
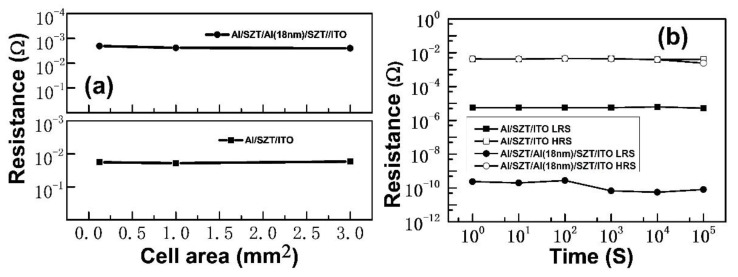
(**a**) LRS current with cell sizes ranging from 0.125 mm^2^ to 3 mm^2^ and (**b**) retention capability of Al/SZT/ITO and Al/SZT/Al (18 nm)/SZT/ITO structures.

**Table 1 nanomaterials-12-04412-t001:** Performance comparison of the SZT-based RRAM devices with reported data.

Insulator Material	Insert Metal	Fabrication of the Insulator	Fabrication of the Insert Metal	Resistance Ratio	Vest (V)	Vreset (V)	Current of HRS (A)	Ref
ZnO	Cu	Sputter	Sputter	10^4^	0.9	−0.6	10^−4^	[11]
TiOx	Pt	Thermal oxidation	--	10^5^	3.7	−0.9	10^−8^	[12]
HfO_2_	Ti	Sputter	Sputter	50	0.7	−1.3	10^−3^	[13]
ZrO_2_	Ti	Electron-beam evaporation	Implant	10^4^	1.3	−0.66	10^−9^	[14]
SZT	--	Sol-gel	Sputter	10^3^	−1.28	2.08	10^−8^	This work
SZT	Al	Sol-gel	Sputter	10^7^	−0.32	1.28	10^−10^	This work

## Data Availability

The data presented in this study are available upon request from the corresponding author.

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
