# Peer review of "Resistive Switching Memory Cell Property Improvement by Al/SrZrTiO3/Al/SrZrTiO3/ITO with Embedded Al Layer"

_nanomaterials, 2022, doi:10.3390/nano12244412_

Round 1
Reviewer 1 Report
1. Introduction part may be modified to highlight previous work related to the SrZrTiO3 (SZT) thin film-based RRAM.
2. Authors should add some references related to the strontium zirconate titanate (SZT) solution preparations.
3. Figure 1(a-c) should be presented more clearly, and the size should be enlarged.
4. For AFM images, authors are requested to modify the figures and clearly present the axis of Figure 2(c).
5. Authors need to check the figure caption of figure 4.
6. It is suggested to add a detail calculations for The trap depth as indicated in page 5.
Author Response
Comments and Suggestions for Authors
- Introduction part may be modified to highlight previous work related to the SrZrTiO3 (SZT) thin film-based RRAM.
Reply: Thank you for your comments. The "Introduction" section of the manuscript has been revised to read as follows. Among various candidates for nonvolatile memories, resistance random-access memory (RRAM) attracts lots of attention owing to its advantages such as simple met-al-insulator-metal structure, fast operation speed, low operation voltage, and low fabrication temperature. Many materials, for example, transition metal oxides (TMOs) including HfO2, TiO2 and NiO, organic material such as composites containing NPs and polyimide, or perovskite oxides such as SrZrO3 and SrTiO3, have been investigated as potential mate-rials for RRAMs. In this study, we focus on the SrTiO3 material because of its high dielectric constant, low dielectric loss, tenability, high breakdown strength, low leakage current, and great film quality [1]. Numerous attempts have been explored to further improve the properties of SrTiO3 based ceramics. Among them, doping was considered as an effective approach for altering their properties [2]. Reproducible resistive switching behaviors have been observed in doped perovskite oxide films [3], such as Mo-doped SrTiO3 [4], Mg-doped SrTiO3 [5], Nb-doped SrTiO3 films [6] and Ni-doped SrTiO3 films [3]. Few studies have indicated that Zr4+ ions in SrTiO3 can stabilize the charge of Ti4+ and suppress the oxygen dissociation [1, 2, 7]. In addition, our previous study showed that the addition of Zr elements can effectively improve the surface morphology of insulators [5]. Also, the sol-gel process has the advantages such as low fabrication temperature, low cost, and adjusting the proportion easily. This process can be applied in many devices [6]. Based on the above advantages and research results, this work, we utilized the Zr element as the doped process, fabricating the SrZrTiO3 (SZT) by sol-gel process for RRAM insulators.
Recently, various inserted metal layers, including Cu, Pt, and Ti in the insulator layer have been shown to improve resistive switching properties of RRAMs [10-13]. However, relatively few studies have been performed regarding the mechanism of the embedded metal-based RRAM devices. In this study, the effects of Al included SZT on the resistive switching properties of SZT thin film for RRAM applications was also investigated. As a comparison, a memory device with a single SZT active layer is also fabricated and characterized.
- Authors should add some references related to the strontium zirconate titanate (SZT) solution preparations.
Reply: Thank you for the valuable suggestions. We have added some literature as a reference for preparing experimental solutions. Although the solutions used in the previous literature are different from this research, they can still be used as a reference for solution preparation.
- Figure 1(a-c) should be presented more clearly, and the size should be enlarged.
Reply: Thank you for your comment. We have replotted the figure 1.
Figure 1. (a) Preparation of the 0.5 M SZT solution. (b) The schematic of SZT based RRAM devices. (c) The schematic of the embedded Al in SZT thin film RRAM devices.
- For AFM images, authors are requested to modify the figures and clearly present the axis of Figure 2(c).
Reply: Thank you for your comment. We have replotted the figure 2(c).
Figure 2. (a) TEM image of the cross-sectional SZT/Al/SZT tri-layered thin films sandwiched between ITO bottom electrode and Al top electrode. (b) The XRD spectrum of SZT thin film. AFM image of (c) the SZT and (d) the SZT/Al/SZT tri-layered thin films.
- Authors need to check the figure caption of figure 4.
Reply: Thank you for your comment. We have replotted the figure 4.
Figure 4. I–V switching curves of the (a) Al/SZT/ITO, (b) Al/SZT/SZT/ITO and (c)-(f) Al/SZT/different thicknesses (5 nm, 18nm, 25nm, and 33nm) of embedded metal –Al /SZT/ITO devices RRAMs.
- It is suggested to add a detail calculations for The trap depth as indicated in page 5.
Reply: According to thermionic emission theory, the saturation current at a voltage of 0.5 V is related to the Schottky barrier heights based on the following equation[R1]:
Isat = AA*T2 exp[−Φt/(kBT)] (1)
where A is the electrode area, A* is the Richardson constant, T is the absolute temperature, kB is the Boltzmann constant, and Φt is the barrier height. Φt,SZT and Φt,SZT/Al/SZT are the barrier height of the SZT and SZT/Al/SZT, respectively. Which implied a higher barrier height of SZT/Al/SZT than expected although complicated interface may exist. The increased barrier height is beneficial to obtain the low HRS current and high ON/OFF ratio.
[R1] F. C. Chiu,“A review on conduction mechanisms in dielectric films,” Advances in Materials Science and Engineering, pp.1-18, 2014. DOI:10.1155/2014/578168

Reviewer 2 Report
Lee K-J, et al. proposed the insertion of Al layer in SrZrTiO3-based memristor.
The manuscript is lack of novelty and there are some serious flaws in the study; which does not suit to the MDPI Nanomaterials standard. Nevertheless, the reviewers think that if the authors could focus more on the chemical and physics phenomenon rather than the performance, this study could offer some novelty.
Hence, I suggest the authors to improve the manuscript based on my comments below:
1. The authors presented the result of the devices made with various Al insertion layer. However, the authors neglect the phenomenon observed in the devices in xnm, 25nm, and 33nm.
2. The device made with 18 nm Al layer shows similar characteristics with the one without. However, if the Al thickness is less or more than 18 nm the I-V becomes analog. Please explain why.
3. In line with previous comment, the authors tend to omit the potential of analog devices. Analog devices can be useful for neuromorphic computing, as suggested in paper [1], regardless their On/Off ratio.
4. If I understand the Fig 4 correctly, it seems to me that a (different) current compliance was used for different devices during the DC sweeping tests. All devices should be measured using the same testing parameter.
5. The O1s XPS spectra analysis from various depth (accros the SZT/Al/SZT structure) could explain the phenomenon observed in Fig. 4.
6. Line 39. I believe the authors intend to say "Recently, it is found that inserting a metal layer such as Cu, Pt, or Ti in the insulator layer could improve...."
7.Line 56. Diameter or area (mm^2)? If it is diameter, why the schematic in Fig.1 shows rectangular shape top electrodes?
8.Line 63. Please explain ultra-thin. Less than 5nm?
9. The authors should explain what equipment and the method/apparatus they used for materials analysis. For example, XRD which geometry? if it is conventional BB apparatus then obviously no signal will be diffracted from thin films. The authors could follow paper [2] on how to explain the methods.
10. In line with previous question, the authors should explain from which depth that the XPS spectra shown in Fig.3a were taken from. Did the authors pre-sputtered the surface of the samples prior to the XPS measurement?
11. Labels in Fig. 2(a) and (c) are unreadable.
12. Line 125. 10 mA is quite high for practical application, hence this contradicts "low power" statement stated in Line 126. Nevertheless, the reviewer agree that -0.32V and 1.28V are low; which could be useful for low-powered devices. Hence, I suggest the authors to calculate the power density to support their low power claims.
13. Fig.5(d) is based on 18 nm Al? This should be identified in the caption.
14. Line 217. Figure numbers are not correct.
15. The authors should study and cite these papers to enhance the quality of manuscripts:
[1]Transformation of digital to analog switching in TaOx-based memristor device for neuromorphic applications (doi: 10.1063/5.0041808)
[2]Flexible nonvolatile resistive memory devices based on SrTiO3 nanosheets and polyvinylpyrrolidone composites (doi: 10.1039/C7TC03481D)
Author Response
Comments and Suggestions for Authors
Lee K-J, et al. proposed the insertion of Al layer in SrZrTiO3-based memristor.
The manuscript is lack of novelty and there are some serious flaws in the study; which does not suit to the MDPI Nanomaterials standard. Nevertheless, the reviewers think that if the authors could focus more on the chemical and physics phenomenon rather than the performance, this study could offer some novelty.
Hence, I suggest the authors to improve the manuscript based on my comments below:
- The authors presented the result of the devices made with various Al insertion layer. However, the authors neglect the phenomenon observed in the devices in xnm, 25nm, and 33nm.
Reply: No significant switching behavior was observed in the embedded structure with different thicknesses (5 nm, 25 nm, 33 nm) of aluminum. It was reported that the HRS current value of memory device increased with increasing the density of oxygen vacancy [1-2]. H. W. Huang et al. reported that the amount of filament paths is proportional to the density of oxygen vacancies [3]. The large oxygen vacancies can form the filament paths easier. However, excess oxygen vacancies will lead the numerous filament paths formed suddenly and further affect the performance and stability of the memory devices in the case of the amorphous thin film. Thus, the HRS currents and reset voltages of different thicknesses (5 nm, 25 nm, 33 nm) memory devices were relatively higher than that of different thicknesses (18 nm) memory device. The XPS results show the trend of oxygen vacancy, and further explain the lower HRS current or higher ON/OFF ratio of different thicknesses (18 nm) RRAM.
[1] F. Shao, Z. L. Lv, Z. Y. Ren, L. P. Zhang, G. L. Zhao, J. Teng, K. K. Meng, X. G. Xu, J. Miao, and Y. Jiang, “High endurance of bipolar resistive switching in a Pt/LaNiO3/Nb:SrZrO3/Cu stack: The role of Cu modulating layer,” Chemical Physics Letters, vol. 739, p.137040, 2020.
[2] W. Y. Chang, Y. T. Ho, T. C. Hsu, F. Chen, M. J. Tsai, and T. B. Wu, “Influence of crystalline constituent on resistive switching properties of TiO2 memory films,” Electrochem. Solid-State Lett., vol. 12, pp. H135-H137, 2009.
[3] Huang, H. W.; Kang, C. F.; Lai, F.; He, J. H.; Lin, S. J.; Chueh, Y. L.; Stability scheme of ZnO-thin film resistive switching memory: influence of defects by controllable oxygen pressure ratio. Nanoscale Research Letters., vol. 8, p. 483, 2013.
- The device made with 18 nm Al layer shows similar characteristics with the one without. However, if the Al thickness is less or more than 18 nm the I-V becomes analog. Please explain why.
Reply: No significant switching behavior was observed in the embedded structure with different thicknesses (5 nm, 25 nm, 33 nm) of aluminum. Therefore, we ultimately chose not to explore elucidation.
- In line with previous comment, the authors tend to omit the potential of analog devices. Analog devices can be useful for neuromorphic computing, as suggested in paper [1], regardless their On/Off ratio.
Reply: Thank you for your comment. In the future, we will consider this suggestion and simulate synapses to explore their possibilities.
- If I understand the Fig 4 correctly, it seems to me that a (different) current compliance was used for different devices during the DC sweeping tests. All devices should be measured using the same testing parameter.
Reply: Thank you for your comment. We have replotted the figure 4. The bipolar and reproducible resistive switching behaviors of the Al/SZT/ITO, Al/SZT/SZT/ITO and Al/SZT/Al (different thickness)/SZT/ITO structures were depicted in Fig. 4(a)-(f). The resistive switching curves were obtained from a positive voltage that reached +4 V and returned to 0 V, followed by a negative voltage that reached −4 V and returned to 0 V with 10 mA compliance current. No forming process is required in this switching behavior.
Figure 4. I–V switching curves of the (a) Al/SZT/ITO, (b) Al/SZT/SZT/ITO and (c)-(f) Al/SZT/different thicknesses (ultra-thin, 18nm, 25nm, and 33nm) of embedded metal –Al /SZT/ITO devices RRAMs.
- The O1s XPS spectra analysis from various depth (accros the SZT/Al/SZT structure) could explain the phenomenon observed in Fig. 4.
Reply: Thank you for your valuable comments. We agree that the O1s XPS spectra is a good way to analyze the mechanisms. However, due to the high affinity of aluminum for oxygen, it is difficult to distinguish whether the oxygen is attracted to the aluminum or the electric field is caused by the trapped electrons. Therefore, we finally chose not to elucidate the mechanism from the XPS spectra.
- Line 39. I believe the authors intend to say "Recently, it is found that inserting a metal layer such as Cu, Pt, or Ti in the insulator layer could improve...."
Reply: Thank you for your valuable comments. We apologize for any inaccuracies in the wording. We have corrected the corresponding paragraph in the article and highlighted it.
In the revision, we have revised the following statements (Page 2): “Recently, various inserted metal layers, including Cu, Pt, and Ti in the insulator layer have been shown to improve resistive switching properties of RRAMs [10-13].”
7.Line 56. Diameter or area (mm^2)? If it is diameter, why the schematic in Fig.1 shows rectangular shape top electrodes?
Reply: Thank you for your valuable comments. We apologize for any inaccuracies in the wording. We have corrected the corresponding paragraph in the article and highlighted it. A 90 nm-thick Al film with an area of 3 mm2 was deposited with a shadow mask by using DC magnetron sputtering as the top electrode (TEs) of the metal-insulator-metal (MIM) structure, while ITO serves as bottom electrode (BEs).
8.Line 63. Please explain ultra-thin. Less than 5nm?
Reply: Thank you for your valuable comments. We have corrected the corresponding paragraph in the article and highlighted it. Embedded ultra-thin Al thickness around 5nm.
- The authors should explain what equipment and the method/apparatus they used for materials analysis. For example, XRD which geometry? if it is conventional BB apparatus then obviously no signal will be diffracted from thin films. The authors could follow paper [2] on how to explain the methods.
Reply: Thank you for your comments. The X-Ray Diffractometer (XRD) spectra were developed via with (BRUKER, D8 DISCOVER SSS Multi Function High Power XRD) by using the copper Kα line with λ = 0.154060 nm.
- In line with previous question, the authors should explain from which depth that the XPS spectra shown in Fig.3a were taken from. Did the authors pre-sputtered the surface of the samples prior to the XPS measurement?
Reply: We apologized for not being clear with description. Figure 3(a) shows the X-ray photoelectron spectroscopy (XPS) patterns obtained after 90 s of Ar+ ion sputtering, thereby representing the bulk layer of SZT thin film.
- Labels in Fig. 2(a) and (c) are unreadable.
Reply: Thank you for your comment. We have replotted the figure 2(c).
Figure 2. (a) TEM image of the cross-sectional SZT/Al/SZT tri-layered thin films sandwiched between ITO bottom electrode and Al top electrode. (b) The XRD spectrum of SZT thin film. AFM image of (c) the SZT and (d) the SZT/Al/SZT tri-layered thin films.
- Line 125. 10 mA is quite high for practical application, hence this contradicts "low power" statement stated in Line 126. Nevertheless, the reviewer agree that -0.32V and 1.28V are low; which could be useful for low-powered devices. Hence, I suggest the authors to calculate the power density to support their low power claims.
Reply: Thank you for your comment. We apologized for not being clear with description. We have corrected the corresponding paragraph in the article and highlighted it. The VSET/VRESET of Al/SZT/ITO is -1.28 V/+2.08 V. The ON/OFF ratio of the devices is around 103. Al/SZT/SZT/ITO RRAM devices were depicted in Fig. 4(b). The set voltage is -2.88 V and the reset voltage is located at +2.48 V. The ON/OFF ratio of the devices is around 10. The discussion resistive switching I-V characteristic of different thicknesses (5 nm, 18nm, 25nm, and 33nm) of embedded metal –Al, as shown in Fig. 4(c)-4(f). Therefore, the observe I-V characteristic of 18nm embedded Al display especially outstanding performance, however the memory margins of other thickness embedded Al layer are not big enough for memory application. More importantly, the VSET/VRESET of Al/SZT/Al (18 nm)/SZT/ITO is -0.32 V/+1.28 V in Fig. 4(d). The ON/OFF ratio of the devices is around 107. No significant switching behavior was observed in the embedded structure with different thicknesses (5 nm, 25 nm, 33 nm) of Al in Fig. 4(c), Fig. 4(e) and Fig.4(f).
- Fig.5(d) is based on 18 nm Al? This should be identified in the caption.
Reply: Thank you for your comments. We have replotted the figure 5.
Figure 5. Double logarithmic plot and linear fitting of the switching I–V curve of (a) Al/SZT/ITO and (b) Al/SZT/Al/SZT/ITO devices RRAMs.
- Line 217. Figure numbers are not correct.
Reply: Thank you for your considerable comments. We apologize for the typos. In the revision, manuscript has been revised.
- The authors should study and cite these papers to enhance the quality of manuscripts:
[1] Transformation of digital to analog switching in TaOx-based memristor device for neuromorphic applications (doi: 10.1063/5.0041808)
[2] Flexible nonvolatile resistive memory devices based on SrTiO3 nanosheets and polyvinylpyrrolidone composites (doi: 10.1039/C7TC03481D)
Reply: Thanks for your valuable comments. We have tried our best to revise the paper according to the comments. Hopefully, all the concerns have been adequately addressed.
Submission Date 11 November 2022
Date of this review 22 Nov 2022 18:40:54

Round 2
Reviewer 2 Report
The authors have addressed all comments.